Questioning the proverb ‘more haste, less speed’: classic versus metabarcoding approaches for the diet study of a remote island endemic gecko

Gil Vanessa 1
Pinho Catarina J. 2 3
Aguiar Carlos A.S. 1
Jardim Carolina 4
Rebelo Rui rmrebelo@fc.ul.pt 1
Vasconcelos Raquel raquel.vasconcelos@cibio.up.pt 2
1 Centre for Ecology, Evolution and Environmental Changes, Faculdade de Ciências da Universidade de Lisboa , Lisboa , Portugal
2 CIBIO, Centro de Investigação em Biodiversidade e Recursos Genéticos, InBIO Laboratório Associado, Universidade do Porto , Vairão , Portugal
3 Departamento de Biologia, Faculdade de Ciências da Universidade do Porto , Porto , Portugal
4 Instituto das Florestas e Conservação da Natureza IP-RAM , Funchal , Madeira , Portugal
Cardoso da Silva José Maria
Electronic publication date: 2020 Jan 2
Publication date: 2020
Volume: 8
Electronic Location ID: e8084
Received 2019 Apr 10; Accepted 2019 Oct 22
Copyright: ©2020 Gil et al.
Copyright year: 2020
Copyright holder: Gil et al.
License: This is an open access article distributed under the terms of the Creative Commons Attribution License, which permits unrestricted use, distribution, reproduction and adaptation in any medium and for any purpose provided that it is properly attributed. For attribution, the original author(s), title, publication source (PeerJ) and either DOI or URL of the article must be cited.
License URL: https://creativecommons.org/licenses/by/4.0/

Keywords: Conservation, Tarentola (boettgeri) bischoffi, Gekkonidae, Selvagens archipelago, Macaronesia, Seasonality, Protected area

Funding: FCT, Fundação para a Ciência e Tecnologia, I.P. SFRH/BPD/79913/2011 The European Social Fund and the Human Potential Operational Program National funds DL57/2016/CP1440/CT0002 FCT projects PEst-OE/BIA/UI0329/2014 Monaco Explorations Research work was supported by FCT, Fundação para a Ciência e Tecnologia, I.P. (SFRH/BPD/79913/2011 fellowship and DL57/2016/CP1440/CT0002 contract to Raquel Vasconcelos), financed by The European Social Fund and the Human Potential Operational Program, POPH/FSE and national funds under the scope of ‘Norma transitória’. Research work was also supported in the frame of FCT projects (PEst-OE/BIA/UI0329/2014 to Rui Rebelo). Field and labwork in 2017 was supported by Monaco Explorations. The funders had no role in study design, data collection and analysis, decision to publish, or preparation of the manuscript.

==============================
Dietary studies can reveal valuable information on how species exploit their habitats and are of particular importance for insular endemics conservation as these species present higher risk of extinction. Reptiles are often neglected in island systems, principally the ones inhabiting remote areas, therefore little is known on their ecological networks. The Selvagens gecko Tarentola (boettgeri) bischoffi, endemic to the remote and integral reserve of Selvagens Archipelago, is classified as Vulnerable by the Portuguese Red Data Book. Little is known about this gecko’s ecology and dietary habits, but it is assumed to be exclusively insectivorous. The diet of the continental Tarentola species was already studied using classical methods. Only two studies have used next-generation sequencing (NGS) techniques for this genus thus far, and very few NGS studies have been employed for reptiles in general. Considering the lack of information on its diet and the conservation interest of the Selvagens gecko, we used morphological and DNA metabarcoding approaches to characterize its diet. The traditional method of morphological identification of prey remains in faecal pellets collected over a longer period was compared with metabarcoding of samples collected during rapid surveys. Molecular results revealed that this species is a generalist, feeding on invertebrate, plant and vertebrate items, whereas the morphological approaches were unable to detect the latter two. These results opened up new questions on the ecological role of the Selvagens gecko that deserves to be further explored, such as the possible predation on seabirds, plant services or trophic competition with the sympatric Madeira lizard Teira dugesii. Metabarcoding identified a greater diversity of dietary items at higher taxonomic resolution, but morphological identification enabled calculation of relative abundances and biomasses of ingested arthropods, and detected a dietary shift on invertebrate preys between seasons. Results of this study highlight the global applicability of rapid metabarcoding surveys for understudied taxa on remote islands that are difficult to access. We recommend using the metabarcoding approach, even if ‘speedy’ sampling only is possible, but we must highlight that disregarding long-term ecological data may lead to ‘hasty’ conclusion.

Introduction

An island is usually considered a natural laboratory due to its geographical nature and to the presence of its uniquely evolved biota (Whittaker et al., 2017). Even though islands generally present a small number of species in relation to mainland systems, high numbers of endemics are known to occur, especially on remote islands (Whittaker & Fernández-Palacios, 2007). These endemic species are more prone to extinction due to the synergy of genetic and demographic factors (Frankham, 1997). Insular systems represent simplified models, ideal for studying ecological networks, as the species inhabit more confined areas which allows more thorough sampling. Studies on such systems are of great importance to ensure accuracy when developing conservation measures (Caujapé-Castells et al., 2010). Comprehending the feeding habits of a species increases the knowledge about the way it exploits its environment. Therefore, dietary studies represent an important topic in conservation. These are essential for gaining insight on species to which information remains scarce (Pérez-Mellado et al., 2011) and contributing to existing studies on the subject (e.g., Neves et al., 2017). An interesting example are insular reptiles, which often vastly differ in their diets when compared to continental congeners (Brock, Donihue & Pafilis, 2014; Sagonas et al., 2015). Some studies show a generalization of diet in island species (Sagonas et al., 2015), while others reveal drastic changes in the trophic niche as a result of the need to adapt to their different insular prey availability (Briggs et al., 2012; Carretero & Cascio, 2010).

Geckos comprise the largest lizard family, with about 2000 species worldwide (inhabiting mainly warm climate regions), and include several iconic examples of island colonization across the three main oceans (Vitt & Caldwell, 2013). Within this family, the genus Tarentola is the most widespread in the Western Mediterranean and presents us with several island endemics in Macaronesia (Rato, Carranza & Harris, 2012; Vasconcelos, Carranza & James Harris, 2010). Several studies based on morphological identification of prey were performed on the diet of the most widespread species, such as Tarentola annularis (Geoffroy-St-Hilaire, 1827) and Tarentola mauritanica (Linnaeus, 1758). In northern Egypt, the former’s diet mainly consists of flying arthropods with some vestiges of plant material (Ibrahim, 2004); however, predation on small mammals has also been reported (Crochet & Renoult, 2008). According to the latest studies specific to the Iberian Peninsula, T. mauritanica dietary habits consist almost exclusively of ground-dwelling arthropods (Gil, Guerrero & Perez-Mellado, 1994; Hódar & Pleguezuelos, 1999; Hódar et al., 2006), contrasting with the patterns found in Rome, Italy, which show mostly flying arthropods, such as Diptera and adult Lepidoptera (Capula & Luiselli, 1994).

Few studies have been performed using next-generation sequencing (NGS) techniques to assess the diet of reptiles, and only two to the best of our knowledge on Tarentola geckos (Pinho et al., 2018; Seguro, 2017). DNA metabarcoding is a non-invasive technique that allows the identification of multiple food items in a species diet through sequencing of standardized DNA fragments (Pompanon et al., 2012). Highly variable DNA regions that enable species-level identification are amplified using universal or group-specific primers which bind to conserved sites across multiple taxa. Metabarcoding is extremely advantageous when studying species dietary habits as it enables the gathering of large datasets from remote areas without requiring the time or effort needed by conventional tools. Compared to traditional methods, metabarcoding maximizes resolution as well as detection of soft, small and inconspicuous prey items, and it is less reliant on taxonomic expertise; ultimately, its use can correct biases in ecological models (Pompanon et al., 2012). In spite of the potential of this approach, it is important to know that there are also methodological implications. One of these being that it only provides species presences and not their proportions in samples (Piñol et al., 2015). The number of reads does not represent the abundance of prey due to differences in prey digestibility (Jarman et al., 2013) and potential biases during PCR amplification (Pompanon et al., 2012). Additionally, the obtained data can be biased during DNA extraction, PCR pooling, sequencing and bioinformatic processing (Pompanon et al., 2012). Nevertheless, as metabarcoding continues to evolve at an accelerated rate, these and other issues are gradually tackled.

The Selvagens gecko, Tarentola (boettgeri) bischoffi (Joger, 1984), is endemic to the remote Selvagens Archipelago (Fig. 1), located about 250 km south of Madeira Island (Oliveira et al., 2005). It occurs in three isolated subpopulations, which correspond to the three largest islands of the archipelago (Rebelo, 2008) (Fig. 1), and is considered Vulnerable by the Portuguese Red Data Book (Oliveira et al., 2005). The closest relatives of this gecko live in two islands of the Canary Archipelago—Gran Canaria, Tarentola boettgeri boettgeri (Steindachner, 1891), and El Hierro, Tarentola boettgeri hierrensis (Joger & Bischoff 1983). The group is related to Tarentola mauritanica populations from North Africa, from which it separated about 17.5 million years ago as a result of an ancient Macaronesian colonization (Carranza et al., 2000).

Previous studies inferred that Selvagens gecko voluntarily eats insects and avoids rotten fruits (Olivera et al., 2010), but studies focused on its feeding habits are still lacking. Considering the poor availability of information on Selvagens gecko diet and its plasticity, we used two approaches in this study to characterize the diet of this endemic species for the first time. We compared the traditional method of morphological identification of prey remains in faecal pellets collected over a longer period of time with metabarcoding of samples collected during rapid surveys. Additionally, we built a reference collection (DNA and morphological), so we could have more robust taxa identification and deduce on food availability between seasons. The results of this study highlight the global applicability of rapid metabarcoding surveys for understudied taxa on remote islands with difficult access.

Figure 1 Study area.

Location of the Selvagens Archipelago in the East Atlantic coast and of the Selvagem Grande Island in the archipelago.

Materials & Methods

Study area

Sampling was carried out on the largest island of the archipelago, Selvagem Grande (Fig. 1), a plateau approx. 120 m a.s.l. which is surrounded by steep cliffs. The climate is characteristically semi-arid, as its low altitude does not favour precipitation (below 200 mm/year). However, occasional winter torrential floods may occur.

The flora of Selvagens Islands is composed of approximately 75 taxa, seven being exclusively endemics. The majority are considered threatened (Sim-Sim et al., 2014). Since the successful eradication of house mouse Mus musculus and European rabbit Oryctolagus cuniculus in 2005 (Olivera et al., 2010), the scarce but steadily recovering vegetation in the plateau is now mainly composed of shrubby sea-blite Suaeda vera Forssk. ex J. F. Gmel with some individuals of the Macaronesian endemic Schizogyne sericea (L.f.) DC. (Penado et al., 2015).

The Selvagens Archipelago, which was classified as an Important Bird Area (IBA) by Birdlife International (Bird Life International, 2019), is one of the most important breeding areas for seabirds in Macaronesia. Nine breeding species occur, and the archipelago plays a key role in the protection of Cory’s shearwater Calonectris borealis (Cory, 1881) by sheltering one of the largest breeding colonies in the world (Granadeiro et al., 2006). There are also two endemic reptiles—the mainly diurnal Madeira lizard Teira dugesii (Milne-Edwards, 1829) and the strictly nocturnal Selvagens gecko Tarentola (boettgeri) bischoffi, our study species. Despite having segregated activity periods, some cases of predation of Madeira lizard on eggs of Selvagens gecko have been reported (Oliveira et al., 2005; Rebelo, 2008). The terrestrial arthropod community on the island is diverse, including 201 taxa (Borges et al., 2008a).

Sampling

Diet studies reliant on prey identification from faecal samples represent only a snapshot of the last ingested meal. Considering that diet composition can differ with prey availability, which fluctuates with seasonal variations, sampling must be conducted over several periods of time to ensure a thorough description of species diet. Taking that into account, two periods were sampled with a classical approach to check if there were differences in Selvagens gecko diet due to season. Sampling took place in intermittent years at the end of summer (6th–15th September 2010 and 10th–11th September 2017) and in late spring (9th–30th May 2011).

Gecko faecal pellets (N = 16 in September 2010, N = 66 in May 2011, and N = 27 in September 2017) were obtained by gently pressing the abdomen of adult individuals (>45 mm SVL; Penado et al., 2015) which were caught by lifting rocks on the plateau during the day. In 2017, and for the metabarcoding analyses, pellets were stored in tubes with 96% ethanol for DNA preservation, and labelled with the respective code of the animal.

The soil arthropod community was sampled with 50 mL pitfall traps, which were left open for 12 h on two occasions per season in 2010/2011 and on one occasion in 2017. To obtain a reference collection of the island’s arthropods for morphological identification, five traps containing water, 70% alcohol and detergent were placed in each of four 1 ha squares at evenly spaced intervals along the island plateau, and left open overnight, in 2010 and 2011. In September 2017, eight traps were placed on two areas of the island for the DNA reference collection of arthropods. These pitfall traps did not contain detergent to prevent DNA degradation. All arthropods were photographed and identified to the family level whenever possible. Specimens collected in 2010 and 2011 were also weighted to obtain an estimate of the average body mass of each taxonomic category. A leg or wing sample was used from the specimens collected in 2017 of each different Operational Taxonomic Unit (OTU) identified by the experts (C. Aguiar; R. Rebelo) to perform DNA extraction. Plant and vertebrate samples (pieces of leafs and dead animals) were also collected in 2017 to build the DNA reference collection, except those already present on GenBank.

The work within the Natural Reserve of Selvagem Grande was carried out with the permission of Parque Natural da Madeira, PNM (Permits in 2010 and 2011, and License nr 09/IFCN/2017). Sampling and protocols were approved by PNM.

Morphological analysis

In 2010 and 2011 pellets were stored dry in plastic tubes, and later dispersed in water and examined with a binocular magnifying glass. The numbers of each prey item in each pellet were estimated from the cephalic capsules, wings (including elytra) and legs, following the criterion of the minimum number (e.g., six formicid legs would count as one individual ant). Prey items were identified to the lowest possible taxonomic level.

Metabarcoding analysis

Arthropod DNA from the reference collection was extracted using saline extraction methods (Carranza et al., 1999) and amplified using a modified version of the IN16STK-1F/IN16STK-1R primers (Tables S1, S2 and S3) targeting the mitochondrial 16S rRNA gene (Kartzinel & Pringle, 2015) to match with diet sequences. The standard COI barcode fragment was amplified using LCO1490/HC02198 primers following PCR conditions described in Folmer et al. (1994). The amplification and sequencing of the latter marker allowed confirmation of dubious taxonomic assignations by comparison with sequences available in the BOLD database (http://boldsystems.org/).

Vertebrate samples from the reference collection were extracted using saline methods and amplified for the V5-loop fragment of the mitochondrial 12S gene using 12sv5F and 12Ssv5R primers (Riaz et al., 2011; Tables S1–S3).

Plants were photographed and identified by experts (M.M. Romeiras), and DNA extracted using DNeasy Plant Mini Kit (Qiagen, Crawley, UK) following some alterations according to Romeiras et al. (2015), and amplified using primer ‘e’ and ‘f’ (Taberlet et al., 1991) targeting the chloroplast intergenic spacer within the trnL (UAA) 3′ exon and trnF (GAA) (Tables S1–S3). DNA from all reference samples was sequenced using Sanger sequencing (310 Applied Biosystem DNA Sequencing Apparatus). Chromatograms were checked manually to detect and correct sequencing errors.

The collected pellets were completely dehydrated in an incubator at 50 °C in order to remove all traces of ethanol. Then, DNA was extracted using the Stool DNA Isolation Kit (Norgen Biotek Corp.Canada), following the manufacturer’s instructions. Three different DNA fragments were chosen to identify the distinct prey groups presumably preyed by the study species. For plants, the g/h primers that target the short P6-loop of chloroplast trnL (UAA) intron were used (Taberlet et al., 2007). For invertebrates and vertebrates, the same IN16STK-1F/IN16STK-1R and 12sv5F/12Ssv5R primers, respectively, were used as described above. To avoid the amplification of T. (boettgeri) bischoffi DNA, a blocking primer (5′-CTCCTCTAGGTTGGTTTGGGACACCGTC (C3 spacer) -3′) was designed, according to previous references by Vestheim & Jarman (2008).

We further modified all primers used for metabarcoding in order to contain Illumina adaptors and a 5-bp individual identification barcode to allow individual identification of each sample.

Succeeding amplification of both pellets and reference collection, library preparation was carried out following the Illumina MiSeq 16S Metagenomic Sequencing Library Preparation protocol (Illumina, 2013) (see Pinho et al., 2018 for details). Samples were pooled per fragment at equimolar concentrations (15 nM). The final pool comprising the three fragments was quantified by qPCR (KAPA Library Quant Kit qPCR Mix, Bio-Rad ThiCycler), diluted to 4 nM, and run on a MiSeq sequencer (Illumina) using a 2 × 150 bp MiSeq Reagent Kit for an expected average of 12,000 paired-end reads per sample.

The sequences were processed using the software package Obitools (https://git.metabarcoding.org/obitools/obitools; see Pinho et al., 2018 for details), using the command illuminapairedend for aligning sequences with quality over 40, the command ngsfilter to assign reads to samples and primers and remove barcodes, and the command obiclean to collapse reads into unique haplotypes. Samples with less than 100 reads and haplotypes representing less than 1% of the total number were excluded. Taxa were assigned by comparing the obtained sequences against GenBank online database using BLAST (Basic Local Alignment Search Tool; https://www.ncbi.nlm.nih.gov/), and lists of species occurring on Selvagens Islands (Borges et al., 2008b; Table S4). Sequences with less than 90% BLAST identity were assigned to class level. Sequences with 90–95% BLAST identity were assigned to the family level. Remaining sequences with 95% BLAST identity or higher were assigned to species or genus level (Table S5). Prey items were identified to the lowest possible taxonomic level (Table S5).

Methods and seasons comparisons

Accumulation curves were built for all three sampling moments considering family level assignments.

The Shannon-Wiener Diversity Index was calculated to characterize the gecko’s diet in the 2010 and 2011 samples and the diversity indices were compared between seasons with t-tests (Zar, 2010). This approach was not used for the 2017 samples as it is not possible to estimate the number of individuals belonging to each prey species using metabarcoding (see ‘Discussion’).

Diet composition was expressed in terms of frequency of occurrence (% FO) for both methods, and as numerical frequency (%N) and percentage of biomass (% B) for the morphological identification only. As the main prey items identified were adult holometabolous insects (see results), for the estimation of the ingested biomass we used the average weight of the exemplars collected in the pitfalls. A relative importance index (RII) was assigned to each taxonomic category from the above metrics as RII = %FO × (%N + %B) (Pinkas, 1971). This calculus was possible only for those species for which we had biomass estimates. All the calculations were performed with Microsoft Excel (2007).

Results

Reference collection

A total of twelve, ten and eleven arthropod families were identified in the samples collected in the pitfalls in September 2010, May 2011 and September 2017, respectively. Four of the invertebrates represent new sequences in GenBank or new records (MN628437, MN628438, and MN628442; Table S4). The abundance or presence of each type of arthropod in each of the three sampling occasions is shown in Table 1. Ants (Formicidae) were the most numerous, with similar abundance in both seasons. All the other families were relatively rare, with the exceptions of Psyllipsocidae in September and a single species of Diptera in May. In 2017, we collected nine plant species and two species of vertebrates for the DNA library (Table 1). We provide 9 new sequences of plants and two of vertebrates known to occur on Selvagens before this study (Plants: MN626687–MN626695; vertebrates: MN628443 and MN628444; Table S4).

Table 1 Taxa collected for the classical and metabarcoding reference collection.

(A) Number of arthropods collected per pitfall in the first two sampling seasons (September 2010 and May 2011). In September 2017 we only recorded the presence of each arthropod category for the DNA reference collection. (B) Plant and vertebrate sampled for the DNA reference collection. NI stands for non-identified preys and • for prey presence.

A				B		
Taxonomic category	09/10	05/11	09/17	Taxonomic category	09/17	
Arachnida				Magnoliopsida		
NI Acari	0.00	2		Apiales		
Pseudoscorpiones				Apiaceae		
Cheliferidae	1	1		Astydamia latifolia	•	
Araneae				Asterales		
Gnaphosidae	7	2		Asteraceae		
Salticidae	–	–	•	Senecio incrassatus	•	
Insecta				Caryophyllales		
Coleoptera				Aizoaceae		
Carabidae	5	2	•	Aizoon canariensis	•	
NI Sp. A	1	0		Mesembryanthemum sp.	•	
Tenebrionidae				Amaranthaceae		
Hegeter latebricola	–	–	•	Chenopodium coronopus	•	
Diptera				Fabales		
Dolichopodidae	2	3		Fabaceae		
Hybotidae	2	0		Lotus glaucus	•	
Limoniidae	1	0		Solanales		
NI Sp. C	6	18		Solanaceae		
Hemiptera				Lycopersicon esculentum	•	
Aphididae	0	1		Solanum nigrum	•	
Cicadellidae	–	–	•	Gentianales		
Hymenoptera				Apocynaceae		
Formicidae	213	147	•	Periploca laevigata	•	
Isopoda						
Porcellionidae	–	–	•			
Lepidoptera						
Cosmopterigidae	–	–	•	Aves		
Pyralidae	–	–	•	Procellariiformes		
Psocoptera				Procellariidae		
Ectopsodidae	1	0		Bulweria bulwerii	•	
Psyllipsocidae	32	1		Calonectris borealis	•	
Zygentoma						
Lepismatidae	2	0	•			
Chilopoda						
Scutigeromorpha						
Scutigeridae						
Scutigera coleoptrata	–	–	•			

Table 2 Composition of the diet of Selvagem gecko Tarentola (boettgeri) bischoffi according to the classic and metabarcoding methods.

The three sampling periods were September 2010, May 2011 and September 2017 (the latter with metabarcoding).

Taxonomic category	%N	%B	RII	%FO	
	09/10	05/11	09/10	05/11	09/10	05/11	09/10	05/11	09/17	
Arthropoda										
Arachnida										
Araneae										
Philodromidae	–	–	–	–	–	–	–	–	3.70	
NI (Acari)	0.72	0.00	0.03	0.00	4.69	0.00	6.25	0.00	–	
Pseudoscorpiones									0.00	
Cheliferidae	1.44	1.08	0.06	0.01	18.77	3.30	12.5	3.03	–	
Insecta										
Blattodea NI	–	–	–	–	–	–	–	–	3.70	
Coleoptera										
Anobidae	2.16	0.54	–	–	–	–	18.75	1.52	–	
Carabidae	3.60	16.76	95.67	98.98	3102.18	4559.1	31.25	39.39	29.63	
Chrysomelidae	–	–	–	–	–	–	–	–	3.70	
Coccinelidae	0.72	0.00	–	–	–	–	6.25	0.00	–	
Curculionidae	0.72	0.00	–	–	–	–	6.25	0.00	–	
Lycidae	–	–	–	–	–	–	–	–	11.11	
Sp. A	0.00	14.59	0.00	0.14	0.00	401.64	0.00	27.27	–	
Sp. B	5.04	11.89	–	–	–	–	43.75	24.24	–	
Scarabaeidae	–	–	–	–	–	–	–	–	7.41	
Staphylinidae	0.72	0.00	–	–	–	–	6.25	0.00	3.70	
Tenebrionidae	1.44	0.00	–	–	–	–	12.50	0.00	11.11	
NI	–	–	–	–	–	–	–	–	11.11	
Diptera										
Cecidomyiidae	–	–	–	–	–	–	–	–	3.70	
Chironomidae	–	–	–	–	–	–	–	–	18.52	
Culicidae	–	–	–	–	–	–	–	–	33.33	
Limoniidae	–	–	–	–	–	–	–	–	3.70	
Muscidae	–	–	–	–	–	–	–	–	11.11	
Psychodidae	–	–	–	–	–	–	–	–	3.70	
Sciaridae	–	–	–	–	–	–	–	–	7.41	
NI	0.00	9.73	0.00	0.49	0.00	263.21	0.00	25.75	3.70	
Hemiptera										
Acanthosomatidae	–	–	–	–	–	–	–	–	7.41	
Aphididae	0.00	1.08	0.00	0.01	0.00	3.30	0.00	3.03	3.70	
Cicadellidae	–	–	–	–	–	–	–	–	7.41	
Flatidae	–	–	–	–	–	–	–	–	14.81	
Lygaeidae	–	–	–	–	–	–	–	–	25.93	
Miridae	–	–	–	–	–	–	–	–	3.70	
Pteromalidae	–	–	–	–	–	–	–	–	3.70	
NI (Homoptera)	1.44	0.00	–	–	–	–	12.50	0.00	–	
NI	–	–	–	–	–	–	–	–	7.41	
Lepidoptera										
Lycaenidae	–	–	–	–	–	–	–	–	3.70	
Noctuidae	–	–	–	–	–	–	–	–	11.11	
Tortricidae	–	–	–	–	–	–	–	–	3.70	
NI	–	–	–	–	–	–	–	–	44.44	
Hymenoptera										
Formicidae	82.01	29.73	4.24	0.34	7007.66	774.63	81.25	25.76	7.41	
Pteromalidae	–	–	–	–	–	–	–	–	3.70	
Psocoptera										
Psyllipsocidae	0.00	2.70	0.00	0.03	0.00	8.26	0.00	3.03	–	
Trogiidae	–	–	–	–	–	–	–	–	25.93	
NI	0.00	11.89	–	–	–	–	0.00	24.24	3.70	
Zygentoma										
Lepismatidae	–	–	–	–	–	–	–	–	25.93	
NI									7.41	
Malacostraca										
NI	–	–	–	–	–	–	–	–	7.41	
Decapoda										
NI	–	–	–	–	–	–	–	–	7.41	
Isopoda										
Porcellionidae	–	–	–	–	–	–	–	–	7.41	
NI	–	–	–	–	–	–	–	–	7.41	
Taxonomic category	%FO	
	09/10	05/11	09/17	
Tracheophyta				
Capparales				
Brassicaceae				
Lobularia	–	–	29.63	
Liliopsida				
Poales				
Poaceae	–	–	40.74	
Magnoliopsida				
Apiales				
Apiaceae	–	–	14.81	
Asterales				
Asteraceae	–	–	29.63	
Caryophyllales				
Aizoaceae	–	–	33.33	
Plumbaginaceae	–	–	44.44	
Amaranthaceae	–	–	40.74	
Cucurbitales				
Cucurbitaceae	–	–	3.70	
Ericales				
Ericaceae	–	–	3.70	
Theaceae	–	–	3.70	
Actinidiaceae	–	–	11.11	
Fabales				
Fabaceae	–	–	14.81	
Lamiales				
Oleaceae	–	–	3.70	
Plantaginaceae	–	–	3.70	
Malvales				
Malvaceae	–	–	3.70	
Rosales				
Moraceae	–	–	3.70	
Rosaceae	–	–	7.41	
Sapindales				
Anacardiaceae	–	–	3.70	
Solanales				
Convolvulaceae	–	–	3.70	
Solanaceae	–	–	14.81	
Zygophyllales				
Zygophyllaceae	–	–	3.70	
Chordata				
Actinopterygii				
Perciformes				
Scombridae	–	–	3.70	
Syngnathiformes				
Centriscidae	–	–	3.70	
Aves				
Charadriiformes	–	–	3.70	
Procellariiformes				
Procellariidae	–	–	11.11	
Reptilia				
Testudines				
Cheloniidae	–	–	3.70	
Notes.

% N percentage number

% B percentage of biomass

RII relative importance index

% FO frequency of occurrence

NI not identified

Gecko’s diet

Diet composition in each of the two seasons and for both methods is expressed in Table 2 and Fig. 2. Using classical methods, a total of 324 specimens from seven orders and 16 different arthropod families were retrieved and identified from the pellets (11 families in September 2010 and 10 families in May 2011). Ants (mainly the common species Monomorium subopacum Smith, 1858) were the most numerous prey in both seasons (Table 2 and Fig. 2); however, their frequency and relative importance index were strikingly lower in May than in September. This shift was due to higher consumption of the much heavier carabids (mainly the common species Hegeter latebricola Wollaston, 1854) in spring. Other beetle species were also more frequently consumed in May, as well as Diptera (Table 2 and Fig. 2). Ants were the most frequent prey in September, having been found in 81.25% of the pellets, whereas in May the most frequent prey was Carabidae (39.4% of the pellets). Prey diversity in the pellets was higher in May than in September (H’ 09/10 = 0.84; H 05/11 = 1.90; t188 = −8,192; P < 0.0001). Considering the percentage of biomass, Carabidae was the most important in both seasons, as the biomass of a single carabid is roughly 500 times that of an ant (as a single carabid weighs roughly 56 mg and ants weight 0.11 mg on average). The higher values of the relative importance index belong to Formicidae in September and to Carabidae in May. No plants or vertebrate OTUs were retrieved from pellets using the classical method (Table 2).

Figure 2 Invertebrate families detected with classical and metabarcoding methods.

Results from September 2010 (in white) and May 2011 (in black) are depicted in the bars on the left and data from September 2017 (in grey) is depicted in the bars on the right. Check Table 2 for further details.

With metabarcoding, after bioinformatic filtering, a coverage of about 5000 sequence reads per sample was obtained. Results revealed invertebrate and plant items presenting almost the same proportion in the samples (77.7% and 74.1% FO, respectively). Vertebrates were also detected in 33% of the samples. With this method, a total of 106 diet items, 62 corresponding to arthropods—19 of them new records for Selvagens, 37 to plants—11 of them new records—and seven to vertebrates were identified (95.1% of sequence reads were assigned taxonomically). For arthropods a total of 12 orders and 29 families were identified. For plants, we were able to identify 16 orders and 21 families, and six orders and six families for vertebrates. The plant family Plumbaginaceae (specifically Limonium papillatum Webb & Berthel, 1891) had the highest frequency of occurrence of the group, and one of the greatest together with non-identified Lepidoptera in the general diet considering all taxonomic groups. For the arthropods, even though non-identified Lepidoptera were more frequent, Culicidae was the arthropod family with a higher incidence in the samples. Regarding vertebrates, Procellariidae (specifically Calonectris borealis) was the family with a higher frequency of occurrence. Other vertebrates were also found in the diet with lower frequencies, such as fishes and reptiles (detected in only one sample).

Taxa accumulation curves using the results of the classic method (Fig. 3) very quickly reached a plateau (after five pellets in both seasons), indicating that even the reduced September sampling effort is probably enough to characterize the species diet. However, using metabarcoding we did not reach that plateau (Fig. 3).

Figure 3 Taxa accumulation curves at family level for the three sampling periods.

Data is presented for September 2010 and 2017 and May 2011. Data resulting from classic morphological studies corresponds to 2010 and 2011 (left y-axis) and for metabarcoding to September 2017 (right y-axis).

Discussion

Our results represent the first data on the dietary habits in nature of the endemic and threatened T. (boettgeri) bischoffi. Looking exclusively into the morphological results, we have concluded that the Selvagens gecko appears to be mostly myrmecophagous at the end of the dry season, shifting to larger prey (especially carabids) during spring. In fact, ants were consistently the most numerous prey in both seasons. The much heavier beetles, which provided the highest biomass, were also consumed in both seasons. In May, although Formicidae continued to dominate the diet of the species, its occurrence was much lower than in September, with an increase of Carabidae, Diptera and other Coleoptera.

Proportion wise, the difference in ant availability between the two seasons was almost non-existent. Therefore, a higher consumption of beetles in May is not explained by a variation in ant availability. The selection of more nutritious prey by the geckos could explain this rise. The increase in consumption of non-ant arthropods, such as Carabidae (which were also the most frequent), some other Coleoptera and Diptera from September to May, coincided with the decrease in ant consumption. Since Carabidae was also the most important prey in terms of biomass in both seasons and presented the higher values for the relative importance index in May, the species seems to show a dietary pattern similar to that described by Hódar & Pleguezuelos (1999) for T. mauritanica. Hódar & Pleguezuelos (1999) showed that between April and July, Lepidoptera and Carabidae larvae and Araneae were the main groups consumed by T. mauritanica, whereas from July to September their presence in the diet decreased with an increase in Homoptera, Coleoptera and Formicidae. This pattern is related to the preference for less sclerotized, highly profitable groups such as larvae in spring and a shift to prey species adapted to drought and food scarcity in summer.

Although evidence exists, it was not possible to prove that the differences in the consumption of prey were due to variations in the food supply. On the other hand, the higher consumption of non-ant arthropods in May coincided with a longer rainy season in the previous months, which agrees with Greenville & Dickman (2005), who showed that flexibility of feeding strategies can be expected in arid environments with a large variation in precipitation. James (1991), in his study regarding Ctenotus species in Australia, observed that the proportion of termites in the diet was higher during drier periods, concluding that termites constitute a good source of food during droughts. The Selvagens gecko seems to be a seasonal ant specialist since this item has always been the most consumed; a possible increase in the availability of other prey will lead to the adoption of a more varied diet, and therefore a more generalist food regime. This supports the theory that many reptiles maintain a flexible diet by opportunistically exploiting diverse food resources when available (Murray & Dickman, 1994).

By adding the metabarcoding results, a different picture emerges: 13 more families of arthropods were recovered comparatively to when using classical methods. Even though the collection of pellet samples for the morphological analysis was conducted during a longer period—sampled during two seasons for approximately three weeks in total, as opposed to only one day of sampling in September 2017—we were able to retrieve more information (16 versus 29 arthropod families). In general, classical diet methods tend to underestimate the frequency of occurrence of prey with parts that are totally digested (Brown, Jarman & Symondson, 2012), such as some soft-bodied arthropods as Zygentoma and Lepidoptera, which were only detected by using DNA-based methods. Classical methods also tend to more easily detect hard-bodied groups such as Coleoptera.

It is important to note that metabarcoding methodologies still do not provide quantitative data on the biomass of prey consumed, rendering impossible to detect of diet shifts such as the ones identified with the morphological analysis. The number of reads of a determined DNA sequence could reflect the amount of food ingested, yet this does not always happen (Acinas et al., 2005; Polz & Cavanaugh, 1998). This issue is related primarily to biological factors, such as the inevitable differences in the number of DNA copies for unit of mass of the ingested prey, or the differential degradation of DNA during digestion, depending on the type of prey (Pompanon et al., 2012). Also, the number of reads can be influenced by technical factors during PCR amplification when the target DNA is exponentially amplified, which is why an accurate marker choice is so important. Additionally, bias can also occur during the extraction of DNA (Martin-Laurent et al., 2001), DNA pooling and sequencing since there is a preference for the amplification of smaller sequences (Porazinska et al., 2010), and during bioinformatic processing (Amend, Seifert & Bruns, 2010). This represents a disadvantage in relation to classic methods which provide more accurate information on the relative abundance of a specific item in the diet, and thus the quantitative importance of certain prey species.

Metabarcoding results revealed that Selvagens gecko does not rely exclusively on arthropods and probably has a more generalist diet, consuming also plant and vertebrate items. This information is important considering the current Vulnerable status of the gecko. Moreover, plants seem to be as important as arthropods in the diet, occurring roughly in the same proportions. Soft and nutrient-rich plant parts, such as nectar, pollen and some fruits are not identifiable in pellets using classic methods as these became imperceptible after the passage through the digestive tract. The small size of this gecko (average snout-vent length circa 6 cm) may prevent ingestion of large seeds or harder plant parts, which may explain the lack of plant material detected using morphological identification of Selvagens gecko pellets. Even though other classic studies detected plants in reptile diets (Ibrahim, 2004; Pietruszka et al., 1986; Rodríguez et al., 2008), those focused on much larger species (Tarentola annularis, Gerrhosaurus skoogi, and Gallotia galloti, respectively) with higher bite force and gape size. In addition, the study of Ibrahim (2004) was based on stomach contents, such as the study of Sadek (1981) on T. dugesii. Therefore, plant parts were probably not completely digested allowing easier detection. Assessing stomach contents of an endemic and threatened species was neither possible nor desirable.

Although metabarcoding detected plants in most pellets, it could be the case that plant DNA might have been detected due to the consumption of phytophagous arthropods. However, given that at least six pellets with plants and no invertebrates were found, we conclude that plants are in fact a primary food item for these geckos. In particular, the species Limonium papillatum, though being rare in the island, appears to be an important diet item. This plant possibly provides some nutritious advantage, such as rich pollen. A similar pattern was observed for another Tarentola endemic from Cabo Verde (Pinho et al., 2018).

Due to the low number of terrestrial predators, it is common for insular reptile species to become diet generalists as they can reach high densities and face higher competition for food (Pérez-Mellado & Corti, 1993). Moreover, the low arthropod availability in these arid systems may force reptiles to expand their dietary range and even become top predators (Miranda, 2017). In this way, an increase in the consumption of plants by island reptiles is usually detected, such as in several Podarcis species of the Mediterranean islands (Pérez-Mellado & Corti, 1993). On islands, reptiles may even play a significant role in seed dispersal and pollination as the number of pollinators is low. This includes some geckos, such as the diurnal Phelsumas and the nocturnal Hoplodactylus (Godínez-Álvarez, 2004). This also might be the case for Selvagens gecko.

Island reptiles may even prey on seabirds, ingesting their eggs or juveniles, or simply their regurgitations, a behaviour observed in other Tarentola species (Alcover & McMinn, 1994; Mateo et al., 2016; Schleich, 1984; Lopes et al., 2019) and also in the sympatric T. dugesii (Matias et al., 2009). In fact, even with the small sampling effort of the metabarcoding approach, it was possible to detect a link between the Selvagens gecko and seven vertebrate OTUs not detected with the classical methods. On the other hand, the single detection of a turtle as a diet item can be a case of secondary predation, by ingestion of faeces of some flying predator that preyed dead turtles elsewhere, or their predators, such as ants that preyed on their remains. As we showed with metabarcoding that our study species is somehow linked to Cory’s shearwater Calonectris borealis, it would be important to study how the gecko interacts with this bird. However, we could not infer the type of trophic relation of this gecko with seabirds. This could vary from predation to commensalism, in this case through secondary ingestion of bird feathers or faeces while scavenging bird regurgitations (as we detected fish DNA in the gecko’s diet). This link could even be due to secondary consumption when feeding on ants, as studies have shown that they can feed extensively on seabirds (Boieiro et al., 2018; Plentovich, Russell & Fejeran, 2018). After considering all evidence we have discarded this last possibility since pellets with confirmed seabird sequences are different from those with confirmed ant sequences. Similarly, we could not distinguish if the low frequency of DNA of other seabirds in the diet of this gecko is related to the inaccessibility of their nests, to the consumption of degraded DNA from their faeces, or both. In addition, it was not possible to detect potential cases of cannibalism mentioned in other works with Tarentola (Mateo et al., 2016), due to the amplification of Selvagens gecko DNA having been prevented by the use of a blocking primer.

Despite the limitations described above, the metabarcoding approach allowed the detection of a much wider range of diet items (56 considering all three groups) and a more accurate taxonomic description of those items than classic methods (e.g., Diptera was only identifiable to the order level with the latter but at family level with DNA-based methods). We do recommend using this approach even if ‘speedy’ sampling only is possible, but we must highlight that disregarding long-term ecological data may lead to ‘hasty’ conclusions. So, the ideal scenario is to complement both approaches and integrate the information they generate, always taking into account the ecological significance of the results.

Considering only the results of the classical method, this species diet is very different from the continental congeners. In an arid zone of south-east Iberian Peninsula, the main groups present in the diet of T. mauritanica were Araneae, Homoptera, Lepidoptera and Carabidae larvae, and Formicidae. Considering prey biomass, the larvae of Lepidoptera and Carabidae dominated the diet, being followed by non-Araneae Arachnida, Araneae and Onyscidae and plants were not very important diet items (Hódar & Pleguezuelos, 1999; Hódar et al., 2006). Similarly, in Central Iberian Peninsula, the most frequent prey were Araneae, Coleoptera, Homoptera, Diptera and Formicidae (Gil, Guerrero & Perez-Mellado, 1994), while in an anthropic environment (historical centre of Rome, Italy) there was a clear predominance of flying groups like Diptera and Lepidoptera (Capula & Luiselli, 1994), captured using a sit-and-wait strategy near artificial light. However, none of the previous studies used metabarcoding, so results can differ even further or new groups can, in fact, be more important than the ones identified. In this way, diets can be more similar than previously thought, considering that some of those groups were in fact detected in the diet of the Selvagens geckos using the metabarcoding approach.

The diet of other Macaronesian Tarentola endemics that resulted from the colonization of the Canaries and Cabo Verde archipelagos is still poorly known. With the remarkable exception of the Cabo Verdean Tarentola gigas, all island endemics are somewhat smaller than continental T. mauritanica (Pleguezuelos, Márquez & Lizana, 2004; Vasconcelos et al., 2012). In Raso islet, Mateo et al. (2016), studied the diet of small-sized T. raziana (similar in size to the Selvagens gecko) and the very large-sized T. gigas using classical methods and found that the first consisted mainly of insects and other ground arthropods and the latter on vertebrates, but also included arthropods and plants. The main taxa in T. raziana’s diet were Coleoptera, followed by Hymenoptera (Mateo et al., 2016), which are also the main preys found in our results using classical methods. The metabarcoding studies of Seguro (2017) on T. raziana and Pinho et al. (2018) on T. gigas revealed, similarly to our results, the importance of plants and arthropods on these species diet and also the presence of vertebrate items, which were previously undetected. Moreover, those studies, equally to ours, described the presence of invertebrate groups never formerly reported. Classic studies on the diet of the sympatric Madeira lizard Teira dugesii found that diet mainly consists of Coleoptera and Formicidae species, even though ingestion of plants and seabird juvenile feathers were also reported (Aguilar, 2016; Rund, 2016; Sadek, 1981). These results are also consistent with ours, confirming wider trophic niches of the reptiles occurring on the Macaronesian Islands. In addition, these results highlight the possible resource competition between the two reptile species on Selvagens, even though empirical observations sustain their spatial segregation and distinct activity patterns (Oliveira et al., 2005; Rebelo, 2008). This, together with the possible predation of T. (boettgeri) bischoffi by T. dugesii (Oliveira et al., 2005; Rebelo, 2008) should be further explored.

Despite these methodological limitations, metabarcoding studies have proven to, with the appropriate procedures, allow successful detection of a range of taxa identified with classic methods and provide different ecological information (Shaw et al., 2016). DNA-based diet studies have been shown to describe the diet profile of a wide range of species with higher resolution and greater efficacy than classical methodologies, e.g., birds (Trevelline et al., 2016), bats (Hope et al., 2014) and sea lions (Hardy et al., 2017). Therefore, DNA metabarcoding is a valuable and revolutionary tool for conservation research and management (Allendorf, Luikart & Aitken, 2012).

Conclusions

In conclusion, allying classical and DNA-based methods provides a more comprehensive description of a species’ diet spectrum, as well as valuable information for the conservation of threatened species. Metabarcoding methods, even when deployed as extremely quick surveys, can deliver holistic results on diet composition, diversity, and ecological networks at relatively low costs (Lopes et al., 2019). In this case, we showed the importance of plants and, to a lesser degree, of vertebrates, for this insular Selvagens gecko. Moreover, metabarcoding provided a large amount of data in a short time without exclusively relying on taxonomic experts. This is important to provide timely information to institutions responsible for species conservation (Ji et al., 2013), especially in areas of difficult access that require urgent conservation actions, as is the case for many remote islands within biodiversity hotspots (Taylor & Harris, 2012; Thomsen & Willerslev, 2015). However, for the accurate management of island endemics, one short visit will not be enough. For instance, thorough DNA sampling in different seasons should be considered and good reference collections should be assembled, as for instance most invertebrate species known to occur on Selvagens have no sequences available on GenBank (see Table S4). Likewise, the detected trophic links should be explored to have a clear picture of the functional relationships of the target species. Based on our results, it is important to clarify whether the links detected between the gecko and Cory’s shearwater Calonectris borealis are related to predation or commensalism, as well as if the Selvagens gecko has any role in the pollination or seed dispersal of some of the plants identified in its diet. Shedding light on these questions will have consequences for the conservation of this gecko species and its ecosystem. Our work will contribute to establishing guidelines for future management of the Selvagens gecko and its habitat.

Supplemental Information

Table S1 Primer used in this study

Fragment length (in base pairs, bp), sequences, and reference of forward and reverse primers are detailed.

Click here for additional data file.

Table S2 Reagents used in the PCRs

The respective volumes for the different primer sets are provided.

Click here for additional data file.

Table S3 PCR conditions used in DNA amplification

The temperature (T), time (t), and number of cycles (NC) for each primer set is detailed.

Click here for additional data file.

Table S4 List of terrestrial taxa known to occur on Selvagens

Taxonomic identifications, bibliographic citations, and availability of sequences on GenBank (at species and genus level) are detailed. Information about if this study provided new sequences of Selvagens taxa is also given.

Click here for additional data file.

Table S5 Raw genetic data

Taxonomic identifications (ID), haplotype sequences and respective number of reads of the diet items recovered from pellets and from the reference collection. The final ID of OTUs correspond to the highest taxonomical classification as possible (see ‘Materials & Methods’ for details). Information about if this study provided new records is also given.

Click here for additional data file.

A special thanks to Bruno Carreira for all the help throughout one of the sampling seasons and Pierre Gilles, Conceição Biscoito and Sandro Correia for help during the sampling in 2017. Thanks to Maria M. Romeiras for plant identification. Also, thanks to the Portuguese Navy and to Yersin’s crew for taking us to the study area, which would have been impossible otherwise.

Additional Information and Declarations

Competing Interests

Author Contributions

Animal Ethics

Field Study Permissions

Data Availability

The authors declare there are no competing interests.

Vanessa Gil and Catarina J. Pinho performed the experiments, analyzed the data, prepared figures and/or tables, authored or reviewed drafts of the paper, approved the final draft.

Carlos A. S. Aguiar analyzed the data, authored or reviewed drafts of the paper, approved the final draft, morphological identification of prey.

Carolina Jardim performed the experiments, prepared figures and/or tables, approved the final draft, permits and logistics.

Rui Rebelo and Raquel Vasconcelos conceived and designed the experiments, prepared figures and/or tables, authored or reviewed drafts of the paper, approved the final draft.

The following information was supplied relating to ethical approvals (i.e., approving body and any reference numbers):

The work within the Natural Reserve of Selvagem Grande was carried out with the permission of Parque Natural da Madeira, PNM (Permits in 2010 and 2011, and License nr 09/IFCN/2017). Sampling and protocols were approved by PNM.

The following information was supplied relating to field study approvals (i.e., approving body and any reference numbers):

The work within the Natural Reserve of Selvagem Grande was carried out with the permission of Parque Natural da Madeira, PNM (Permits in 2010 and 2011, and License nr 09/IFCN/2017). Sampling and protocols were approved by PNM.

The following information was supplied regarding data availability:

‘SAMN’ accession numbers are in the biosample database (ex. https://www.ncbi.nlm.nih.gov/biosample/SAMN13115936/) and the others are in the nucleotide database (ex. https://www.ncbi.nlm.nih.gov/nuccore/MN628436).

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
