# Peer review of "Questioning the proverb ‘more haste, less speed’: classic versus metabarcoding approaches for the diet study of a remote island endemic gecko"

_PeerJ, doi:10.7717/peerj.8084_

## Round 0.1 · original submission · Major Revisions

I received comments of two reviewers. All of them see merits in your paper, but they think your manuscript needs a major revision to be accepted. Major concerns are: (1) language, as the manuscript requires a through grammar and style review by a native speaker of English to improve clarity and readability; (2) methods, as you should clarify various aspects of their DNA metabarcoding workflow to ensure reliability and reproducibility of their results; (3) datasets, as you should deposit the raw sequence reads on the NCBI Sequence Read Archive as well as deposit all bioinformatics/R scripts along with associated data in a data repository (e.g., DRYAD, GitHub with archival on Zenodo). These issues must be addressed to allow the reader to fully assess the authors’ workflow, results, and discussion. In general, I agree with most of the several excellent suggestions made by the reviewers. Thus, If you decide to revise the work, please follow ALL recommendations and submit a list of changes or a rebuttal against each point which is being raised when you submit the revised manuscript.

·

Basic reporting

Your study is important as it provides the first results on the diet of an endemic species. The main focus seems to be on the comparison of the two techniques so there should be a more clear-cut conclusion as why the question in your manuscript title is posed and what are your views on it. The discussion could include a rational on when to use one technique or the other or on how to use them complementary. I would also like to see some more discussion of the ecological significance of the results.
Other studies have identified plant material in lizard faecal pellets using a morphological technique (e.g. Pietruzka et al., 1986; Rodríguez et al., 2008; Ibrahim, 2004). What do you think is the reason for you not having found plant material in the faecal samples using the classical approach, despite their being an important component of Selvagens gecko diet?
Sadek (1981) conducted a diet study on a different lizard species in Selvagens islands and found considerable amounts of plant material in the stomachs he analysed. Maybe you could quote this study and discuss ecological interactions and potential resource competition between the two species?
Another study that could be worthwhile quoting is Matias et al., 2009, who also found that the Madeiran wall lizard in Selvagens is able to prey on seabirds.
The English language could be improved throughout the manuscript. Some sentences are too long and a bit confusing. Some examples where the language could be improved include lines 7 & 8, 51 & 52, 95 to 101, 286 to 288.
When quoting references throughout the manuscript please follow PeerJ instructions and add a “,” between the author and the year. For example, in line 3 write (Whittaker et al., 2017), instead of (Whittaker et al. 2017).

ABSTRACT
“Reptiles are often neglected in islands systems” - Should be “island” (singular not plural)
“remote and integral reserves of Selvagens Archipelago “ – Should be “reserve” (singular not plural)
“revealed that this species is a generalist eater” – consider replacing “generalist eater” by “generalist feeder”.
“The traditional method of morphological identification of prey remains in faecal pellets collected over a long period was compared with” – Instead of saying “over a long period” I would say “over a longer period”. That is a longer period than the one day sampling for NGS, but not really an extended period covering the entire year/seasons spectrum.
“feeding on invertebrate, plant and even vertebrate items” – please, remove the word “even”. It is not unusual that lizards feed on vertebrates.
“Results of this study are useful to highlight the applicability of rapid surveys on remote islands of difficult access around the world.“ – not really new, the same could be concluded from the results of Pinho et al., 2018.
The abstract is too general; you should include some more of your results, namely the dietary shift between late Spring and end of Summer. Please also discuss the ecological significance of your results and provide some feedback on the question in the title and what you consider the best approach for a future dietary study.
INTRODUCTION
Lines 5 to 10 – All the three sentences begin with the word “These”. Please try to make the writing more fluid.
Lines 11 to 13 – Diet studies are not only essential as a one-off experiment for unstudied species. Diet studies are also especially important as a line of research in the long run and over several years to allow for comparisons and detect environmental changes.
Line 14 – Please follow PEERJ instructions for quoting an article with three authors. It should be (Brock, Donihue & Pafilis, 2014), instead of (Brock et al. 2014).
Line 22 - Please follow PEERJ instructions for quoting an article with three authors.
Lines 31 and 32 - Few studies were performed using next-generation sequencing (NGS) techniques to assess the diet of reptiles, but none (to our best knowledge) on Tarentola geckos (Seguro 2017). 

This statement is incorrect. Not only have Tarentola species diets been studied by NGS, but you include the study in your references list – Pinho et al., 2018. Please remove the sentence and make sure to quote Pinho et al. 2018 in the Introduction.
Lines 37 and 38 – Metabarcoding can be advantageous for diet studies, mainly for insular species from remote areas, as with less effort and time it is possible to obtain a large amount of data.
I don´t agree, metabarcoding it is useful for a broad-spectrum of study sites, not mainly in remote islands. Additionally, if you are comparing effort and time of different methods, you should also include cost in the equation.

Line 41 – You have referred to the advantages of metbarcoding for studying diet, but please also add some references to the limitations of the method.
Line 43 – Please add the word “located” before “about 250 km”
Line 44 and 45 – You could add a reference to the figure, where the name of the three islets that hold Selvagens gecko populations can be seen, e.g. add (see Fig 1)

Line 55 – As suggested for the abstract I would rather read “over a longer period of time” than “over a long period of time”.
Line 51 to 57 – Confusing at the moment, the research question and aim need to be more clearly defined. The most important aspect of the study is not to assess the applicability of NGS methods in remote areas and this is not a good punch line for the introduction. This has been done extensively for several taxa, such as seagrass, birds, mammals and reptiles. In my view the most important aspect is that you are characterising for the first time the diet of an endemic species. You should also describe that you prepared a reference collection (DNA and morphological), as this is an important part of the study and has no mention in the aims. Did you have any intention of using the data collected with the pitfall traps to infer (or at least speculate on) food availability in the two seasons studied (Spring and Late Summer)? If so you should state in the in aims.

Maybe you could list your funding sources after the acknowledgments?
Table 1 - Please format the alignment of the values in column 2 (under 9/10).
Table 2 – Please include the English name - Selvagens guecko - in the legend of the table.
Figure 2 – Title is very basic, please expand.
Supplemental material
Please include the titles for Tables S1, S2, S3 and S4.
Table S1 – Taberlet et al. 2007 is missing from the References list. The sequences of primer 12sv5 are reversed. “TTAGATACCCCACTATGC” is for primer R and not F. Similarly, the sequence given for primer F is in fact for primer R.
Table S2 – Please indicate the units for the reagents in the table.
Table S4 - Supplemental excel file – Please correct line 6. Brassicales is an order, not a classe. Brassicaceae is a family, not an order. Lobularia is a genus, not a family.
Thank you for providing the raw data. You don´t include a table with the sequences of the arthropods, plants and vertebrates sampled for the DNA collection. I supposed you have made them available at an online reference collection / GenBank? If so please include the accession numbers in the manuscript. If not, maybe you could include it as Table S5?

Experimental design

The focus of the study is well within the aims and scope of PeerJ.

The research question is relevant and meaningful but could be better defined.

The methods are sufficiently detailed but should be organised in a more rational/coherent form.

MATERIALS and METHODS
Lines 62 & 63 – No need to repeat the years when sampling was conducted. You mention it below in lines 83 and 84, under the “Data collection and analysis”, so please do not have it there too.
Line 67 – Classified as threaten by whom?
Lines 72 and 73 – quote appropriate reference
Line 74 - remove “for which these islands embody one of the last sanctuaries in the world“

Line 77 – please provide the English name of Teira dugesii
Lines 77 and 78 – You refer to the occurrence of reptiles in a very casual manner. You could state here that one of the species will be the object of your study. Further along in the discussion it would be interesting to see some comparison of the two species.
Lines 81 to 137 – This section needs improvement and a more coherent description. At the moment, it is all mixed, including the description of the two techniques. Lines 110 to 114, describing how and when the samples were collected, should came at the beginning. Only after should you refer to the reference collection justifying why it was needed. And finally, you should describe the methods, one after the other. It would even be better if you split it in different sub-sections, e.g. sampling, morphological analysis, NGS or Metabarcoding and Statistical analysis.
Line 83 and 84 – Simplify, maybe you could say “and late spring of 2011, from 9 to 30 of May.”, instead of “and late spring of just one year, from 9 to 30 of May 2011.”
Line 96 – Should be “tables” instead of “table”
Lines 97 and 124 – Please add the word “gene” after 16S rRNA.
Line 95 to 101 – Sentence too long, please divide in two or three.
Lines 94 and 106 – Please indicate the name of the experts that identified the OTU’s. Or at least list them in the acknowledgements.
Line 105 – should be tables in plural
Line 108 – Please add the word “intron” after (UAA).
Line 111 - Please add the words “the abdomen” after “gently pressing”.
Line 122 – Taberlet et al., 2007 is not in the references list.
Line 135 – Here you should also quote Borges et al., 2008a
Line 139 – Please indicate which software you used for the statistic calculations
Line 152 - Is there no permit reference # for 2010 and 2011?

Validity of the findings

RESULTS
It would be clearer to have two sub-section on the results, one on the reference collection and one on the gecko diet.
Lines 162 and 163 – Please simplify…For example, say “ In 2017, we collected nine species of plants and two species of vertebrates for the DNA library (Table 1).”, instead of “For plants and vertebrates, we collected in 2017 a total of nine and two species, respectively (Table 1).” 

Line 164 – “two sampling seasons”. If you are referring to Spring and Late Summer than just say “the two seasons”. If you are referring to the three years of sampling then say “three sampling periods/years”. The way it is now - “two sampling seasons” - adds unnecessary confusion.
Line 169 – The relative importance in terms of biomass is not so different between September 2010 and May 2011.
Line 193 to 196 –If you only need five or more pellets to characterize the diet using a morphological approach than you can probably do it in one day, while if 27 pellets are not enough to characterize the diet using DNA techniques, you probably need more time. So how does this affect the idea you put forward before that you need a long time period for morphological studies and a short time period for DNA studies?
DISCUSSION
Line 200 – “diet of the endemic” instead of “diet of endemic”
Line 200 to 206 – This would be better in the introduction or in the methods to justify your sampling periods.
Lines 213 to 216 – This sentence is unclear and too long. Please rephrase it.
Line 232 – species should not be in italic
Line 234 and 234, Line 264 – please use the English name instead of the Latin one.
Line 240 – “In fact” implies connection with the previous statement so it shouldn´t be the start of a new paragraph.
Line 243 – The sampling period was one and a half week in 2010, not three weeks.
Line 246 - Please follow PEERJ instructions for quoting an article with three authors.
Line 259 – should be “DNA pooling and sequencing” instead of “DNA pooling, sequencing”
Line 261 – Please follow PEERJ instructions for quoting an article with three authors.
Line 270 – 6 pellets out of the total of 27 pellets analysed is not so much. Please give some more details into why you think secondary consumption is not the main cause for the large proportion of plants you found in the Selvagens gecko diet.
Line 280 – please add English name or sp. in front of Podarcis if you are referring to a group of species.
Line 291 – it should be “their remains” instead of “there remains”
Line 313 - Please follow PEERJ instructions for quoting an article with three authors.
Line 320 – What do you mean by “natural diet”?
Line 323 - Please follow PEERJ instructions for quoting an article with three authors.
Line 347 – By allying do you mean that they should be conducted simultaneously? Explain better what you mean by “allying” in practical terms.
Line 349 – Please replace the word “allied” by connected/linked or coupled to avoid using allying and allied so close together in the text.
Lines 351 and 352 – Are you confident of the importance of plants in the diet?
Line 359 – You suggest DNA sampling in different seasons; are you confident that this will enable you to detect changes in the diet given the limitation in the DNA method you discussed before?
Lines 361 to 364 – Several studies have shown that ants can feed extensively on seabirds. See, for example Boieiro et al., 2018 and Plentovich et al., 2018. How do you know that the seabird items you found in the diet of the Selvagens gecko are not due to secondary consumption? You should at least discuss the possibility.
Boieiro, M., Fagundes, A.I, Gouveia, C., Ramos, J.A. & Menezes, D. (2018) Small but fierce: invasive ants kill Barolo Shearwater (Puffinus lherminieri baroli) nestling in Cima islet (Porto Santo, Madeira Archipelago). AIRO - Portuguese Society for the Study of Birds, 25:44-50.
Plentovich, S., Russell, T. & Fejeran, C. C. 2018. Yellow crazy ants (Anoplolepis gracilipes) reduce numbers and impede development of a burrow-nesting seabird. Biological Invasions 20: 77-86.

Reviewer 2 ·

Basic reporting

The grammar needs to be improved substantially for clarity and readability of the manuscript. I have offered suggestions on this for the Introduction, but this was too time-consuming to do for the entire manuscript.

References are sufficient but more references to recent DNA metabarcoding studies would be welcome. Sufficient background/context was provided.

The article structure could be improved with clearer paragraph breaks and further subdivision of the 'Data collection and analysis' section. Table 2 is very long and difficult to absorb the information. I feel these data could be more easily interpreted as a figure(s).

The authors need to clarify various aspects of their DNA metabarcoding workflow to ensure reliability and reproducibility of their results, namely sample collection, reference database curation, sequencing, bioinformatics, and data accessibility. In particular, I am quite concerned about the lack of detail provided on sequencing (e.g. reagent kit used, number of sequence reads passing each stage of bioinformatic filtering, percentage of sequence reads that were taxonomically assigned), bioinformatic processing, and data accessibility. The raw sequence reads should have been deposited on the NCBI Sequence Read Archive, and bioinformatics/R scripts along with associated data should have been deposited in a data repository (e.g. DRYAD, GitHub with archival on Zenodo). These issues must be addressed to allow the reader to fully assess the authors’ workflow, results and discussion.

Experimental design

I cannot confirm whether the experimental design has been performed to a high technical standard due to details on DNA metabarcoding that were not provided in the manuscript or supplementary information. I do not feel that the methods have been described with sufficient detail that would allow them to be replicated.

Validity of the findings

I cannot confirm whether the underlying data are robust, statistically sound and controlled as these do not appear to have been archived in a data repository. I would like more discussion on how the findings could be incorporated into current/future conservation and management strategies for this gecko species.

Additional comments

General comments

This is an interesting manuscript that uses DNA metabarcoding in conjunction with morphological identification of prey items to assess diet of a remote island endemic gecko. The study nicely shows that DNA metabarcoding can complement and enhance traditional dietary assessments to improve understanding of resource exploitation and trophic adaptation in threatened species.

Metabarcoding revealed the gecko was a generalist, with a broader diet than found previously, and identified more diversity at higher taxonomic resolution than morphological identification of prey remains. The findings should improve conservation and management of this gecko species, although I would like more discussion on how the findings could be incorporated into current/future conservation and management strategies. Nonetheless, the study demonstrates that DNA metabarcoding could be applied to diet of other lizards in continental or island settings. The experimental design, sampling strategy and statistical analyses are largely sound.

However, I have two primary concerns which when resolved would substantially improve the quality of the manuscript. Firstly, the authors need to clarify various aspects of their DNA metabarcoding workflow to ensure reliability and reproducibility of their results, namely sample collection, reference database curation, sequencing, bioinformatics, and data accessibility. In particular, I am quite concerned about the lack of detail provided on sequencing (e.g. reagent kit used, number of sequence reads passing each stage of bioinformatic filtering, percentage of sequence reads that were taxonomically assigned), bioinformatic processing (e.g. OBItools programs and parameters used), and data accessibility. The raw sequence reads should have been deposited on the NCBI Sequence Read Archive, and bioinformatics/R scripts along with associated data should have been deposited in a data repository (e.g. DRYAD, GitHub with archival on Zenodo). These issues must be addressed to allow the reader to fully assess the authors’ workflow, results and discussion.

Furthermore, grammar needs to be improved substantially for clarity and readability of the manuscript. I have offered suggestions on this for the Introduction, but this was too time-consuming to do for the entire manuscript. I have a number of minor comments to help improve the clarity of the manuscript, which I have detailed in the specific comments to the authors below.



Specific comments

Abstract

Line 3: Change ‘islands’ to ‘island’.

Lines 4 & 6: Change ‘on’ to ‘about’.

Line 7: Change ‘supposed to be exclusively insectivorous’ to ‘but they are assumed to be exclusive insectivores’.

Lines 8-10: Change ‘However, using next-generation sequencing (NGS) techniques only one study was performed thus far for this genus and very few for reptiles in general’ to ‘only one study has used next-generation sequencing (NGS) techniques for this genus thus far, and very few NGS studies have been employed for reptiles in general’.

Line 14: Change ‘with metabarcoding methods associated with rapid sampling surveys’ to ‘with metabarcoding of samples collected during rapid surveys’.

Line 15: Remove ‘eater’.

Line 16: Change ‘even though’ to ‘whereas’.

Line 17-20: Change ‘This method identified a higher diversity of dietary items and with high taxonomic resolution. On the other hand, with the traditional method it was possible to calculate relative abundances and biomasses of the ingested arthropods, a parameter that was not possible to measure with metabarcoding’ to ‘Metabarcoding identified a greater diversity of dietary items at high taxonomic resolution, but morphological identification enabled calculation of relative abundances and biomasses of ingested arthropods’.

Line 20-22: Change ‘Results of this study are useful to highlight the applicability of rapid surveys on remote islands of difficult access around the world’ to ‘Results of this study highlight the global applicability of rapid metabarcoding surveys for understudied taxa on remote islands that are difficult to access’.


Main text

Line 3: Change ‘present’ to ‘host’.

Line 8: Change ‘being’ to ‘that are’.

Line 9: The authors should clarify what “These studies” are. Do they mean studies of endemic taxa on remote islands?

Line 12: Change ‘continuing to be’ to ‘and are’, and ‘for which there is’ to ‘with’.

Line 15: Remove ‘the’.

Lines 17-18: Paragraph breaks should be indicated with an indentation or new line here and throughout the rest of the manuscript.

Lines 18-19: Change ‘worldwide, inhabiting mainly warm climate regions,’ to ‘worldwide (mainly in warm climate regions),’. Change ‘including’ to ‘include’.

Lines 22-23: Remove ‘classical’ and commas before ‘based’ and after ‘prey’.

Line 25: Change ‘The former’s diet, in northern Egypt’ to ‘In northern Egypt, the former’s diet’.

Line 26: Remove ‘also being reported’.

Line 28: Change ‘ ground dwelling’ to ‘ground-dwelling’.

Lines 29-30: Change ‘while in the historical centre of Rome, Italy, it is composed mainly of flying arthropods, such as Diptera and adult Lepidoptera’ to ‘but is composed mainly of flying arthropods, such as Diptera and Lepidoptera, in historical Rome, Italy’.

Line 31: Change ‘were’ to ‘have been’.

Line 32: Change ‘but none (to our best knowledge)’ to ‘and none to the best of our knowledge’.

Line 33: Change ‘Metabarcoding’ to ‘DNA metabarcoding’. It is important to distinguish DNA from environmental DNA in the context of metabarcoding.

Line 34: Insert ‘of’ before ‘standardized’.

Line 35: Remove ‘the’ and change ‘general’ to ‘universal’.

Lines 35-36: This is too repetitive of the previous sentence. Change ‘This method is based on the mass-amplification of DNA using general or group-specific primers, followed by the cloning and sequencing of amplicons for individual taxa identification’ to ‘Highly variable DNA regions that enable species-level identification are amplified using universal or group-specific primers which bind to conserved sites across multiple taxa’.

Line 37: Change ‘mainly for insular species from remote areas, as with less effort and time it is possible to obtain a large amount of data’ to ‘especially for insular species from remote areas as less effort and time is required to obtain large datasets’.

Line 38: Change ‘Additionally, in relation to classic methods’ to ‘ Compared with traditional methods,’.

Line 39: Change ‘resolution, and’ to ‘resolution as well as’.

Line 40: Change ‘; ultimately’ to ‘. Ultimately,’.

Lines 43-44: Change ‘, occurring’ to ‘. It occurs’.

Line 45: Change ‘. It is considered’ to ‘, and is considered’.

Line 48: Change ‘with’ to ‘ to’.

Line 49: Insert ‘it’ before ‘separated’.

Line 53: Move ‘in this study’ to come after ‘approaches’.

Lines 55-56: Change ‘with metabarcoding methods associated with rapid sampling surveys’ to ‘with metabarcoding of samples collected during rapid surveys’.

Lines 56-57: Change ‘Results of this study are useful to highlight the applicability of rapid surveys on remote islands of difficult access around the world’ to ‘Results of this study highlight the global applicability of rapid metabarcoding surveys for understudied taxa on remote islands that are difficult to access’.

Line 79: Why is taxa italicised?

Line 86: Why did the number of occasions differ?

Line 87: Please state the percentage of alchohol used, and whether absolute ethanol was used or some other variation.

Line 88: Use of “scattered” seems vague. It would be better to say “randomly selected”, but even so I wonder why the authors did not conduct more systematic trapping transects?

Lines 102-103: The authors should really provide a summary of what species present on the island were represented on GenBank/BOLD, what species were missing, and what species were barcoded in their study.

Line 103: Sanger sequenced where? How were the Sanger sequences validated to ensure they was no sequencing error/misidentification of source DNA?

Lines 104-105: I’m curious as to why the authors chose a saline method instead of a commercial kit, such as the Qiagen DNeasy Blood and Tissue Kit for vertebrate DNA extractions?

Line 116: Stored at what temperature?

Line 118: How long were samples left in the incubator? The standard method of removing ethanol is generally to pour the majority out and leave the pellet to air dry in a fume hood at room temperature.

Lines 123-124: I wonder why the authors did not use COI metabarcoding primer set as well as or instead of 16S rRNA given the amount of invertebrate reference sequences available for the COI region in addition to COI sequences the authors generated themselves.

Line 127: 8ul of blocking primer (10uM) seems excessive – did the authors optimise PCR conditions?

Lines 130-132: What MiSeq reagent kit was used for sequencing? Were COI, 12S and 16S amplicons prepared as one library on a single sequencing run, or were separate sequencing runs performed for each marker? This information should be reported to allow readers to assess whether results were influenced by sequencing depth and coverage.

Line 133: Although OBItools is a standard bioinformatics pipeline to process metabarcoding data, the programs employed across studies vary drastically. I would like the authors to supply more detail on which OBItools programs they used, either in the main text or supplementary information.

Line 134: GenBank is a public reference database – not a method of taxonomic assignment. I suspect that the authors used BLAST for taxonomic assignment. In this case, the BLAST identity (%) and query alignment to reference sequences (%) should be reported.

Line 351: I think the authors need to be careful about inferring ecological networks from their results as they did not actually construct species or trophic networks.


Table 2: It is difficult to compare the morphological and metabarcoding results in table format. A figure (e.g. venn diagram, upset plot, barplot) would be much more intuitive.

---

## Round 0.2 · Minor Revisions

Thank you for the new version of your manuscript. I can see a significant improvement compared to the last version. However, one of the reviewers, who is a global expert in the subject, has detected several small problems in your paper. These problems need to be fixed before I can accept your article for publication. The reviewer provided a very detailed set of recommendations that you need to follow to strengthen and improve the clarity of your arguments.

I look forward to receiving a new version of your paper, including all suggestions by the reviewer.

Reviewer 2 ·

Basic reporting

Grammar in the Abstract and Introduction is much improved, but some text can be more concise. The use of third person when referring to methods or species should also be minimised and names used for clarity throughout the manuscript. I have highlighted these instances in my specific comments to the authors. There are some outstanding grammar issues within the Methods and Discussion that need to be addressed as detailed in my specific comments.

The additional references are appropriate and the article structure is improved. The inclusion of Figure 2 makes it easier to interpret the dietary data and I accept the retention of Table 2. I am satisfied with the authors’ efforts to clarify various aspects of their DNA metabarcoding workflow to ensure reliability and reproducibility of their results, namely sample collection, reference database curation, sequencing, bioinformatics, and data accessibility.

Experimental design

The reorganisation of the methods section and the extra detail that has been added is sufficient. However, I think justification of primer choice should be provided if in silico analysis was not undertaken. If in silico analysis was performed, these results should be reported.

Validity of the findings

I accept that the authors will deposit their data and code on appropriate data repositories upon acceptance. The added text to the discussion on future management strategies for this gecko species is adequate. I accept the validity of the findings bar the issue of in silico analysis.

Additional comments

Abstract

Line 3: Change ‘they’ to ‘these species’.

Line 7: Insert ‘by the Portugese Red Data Book’ after ‘Vulnerable’. Change ‘their’ to ‘this gecko’s’ and ‘they are’ to ‘it is’.

Line 9: Insert ‘have’ before ‘used’.

Line 16: Insert ‘up’ after ‘opened’.

Line 19: Provide binomial name for the Madeira lizard and change ‘high’ to ‘higher’.

Line 21: Change ‘detection of’ to ‘detected’.

Line 23: Remove ‘do’.

Line 24: Insert ‘,’ after ‘approach’ and move ‘only’ after ‘sampling’.


Introduction

Line 28: Change ‘unique’ to ‘uniquely’, ‘it’ to ‘islands’ and ‘presents’ to ‘present’.

Line 30: Change ‘in’ to ‘on’.

Line 33: Change ‘allow sampling more thoroughly’ to ‘allows more thorough sampling’.

Line 38: Remove ‘both’ and change ‘to’ to ‘for’.

Line 39: Change ‘to contribute’ to ‘contributing to’ and remove ‘already’.

Line 42: Change ‘islands’ to ‘island species’ or ‘island reptiles’ and ‘the’ to ‘their’.

Line 47: Change ‘in’ to ‘across’.

Line 54: Insert ‘,’ after ‘however’.

Line 56: Change ‘specifically’ to ‘specific’. ‘Dietary habits’ of which Tarentola species?

Line 58: Change ‘contrastingly’ to ‘contrasting with’.

Line 68: Change ‘habits,’ to ‘habits as’.

Line 69: Change ‘that would have to be invested without the existence of this tool’ to ‘needed by conventional tools’.

Line 70: Change ‘this technique’ to ‘metabarcoding’

Line 71: Insert ‘,’ after ‘ultimately’.

Line 73-74: Change ‘. One of them would be that it only provides the species presences, and not their proportions in the samples’ to ‘, one of these being that it only provides species presences and not their proportions in samples’.

Line 79: Change ‘metabarcoding techniques evolve in accelerated rates’ to ‘metabarcoding continues to evolve at an accelerated rate’.

Line 83: Change ‘South’ to ‘south’. Direction is only capitalised when referring to the name a place, e.g. South America.

Line 89: Change ‘17,5’ to ’17.5’.

Line 92: Change ‘referred’ to ‘inferred’ and ‘avoid’ to ‘avoids’.

Line 94: Change ‘small’ to ‘poor’ and ‘its diet’ to ‘Selvagens gecko diet’.

Line 95: Move ‘for the first time’ to the end of the sentence.

Line 99: Change ‘to infer on’ to ‘deduce’.


Methods

Line 107: Change ‘in’ to ‘on.

Line 113: ‘considered threatened’ by who? The Portugese Red Data Book? IUCN? Government?

Line 114: Provide binomial names for house mouse and rabbit.

Line 119: Insert ‘an’ after ‘as’.

Lines 121-123: Change ‘and specifically for the protection of Cory’s shearwater Calonectris borealis (Cory, 1881) the archipelago plays a key role, sheltering one of the largest breeding colonies in the world’ to ‘and the archipelago plays a key role in the protection of Cory’s shearwater Calonectris borealis (Cory, 1881) by sheltering one of the largest breeding colonies in the world’.

Lines 125 -126: Change ‘object’ to ‘species’. Change ‘Despite the segregation in the period of activity of the two,’ to ‘Despite having segregated activity periods,’.

Line 128: Change ‘in the Island’ to ‘on the island’.

Line 135: Remove ‘our’.

Line 136-137: Change ‘in’ to ‘with’ and ‘of some groups regarding seasonal variation’ to ‘in Selvagens gecko diet due to season’

Lines 137-139: This is very convoluted. I suggest changing to ‘Sampling took place in intermittent years at the end of summer (6th – 15th September 2010 and 10th – 11th September 2017) and in late spring (9th – 30th May 2011)’.

Line 141: Change ‘,’ to ‘;’ or vice versa, but separation of list items should be consistent.

Line 144: Remove ‘,’ after ‘ethanol’.

Lines 148-151: Reword as ‘To obtain a reference collection of the island’s arthropods for morphological identification, five traps containing water, 70% alcohol and detergent were placed in each of four 1 ha squares at evenly spaced intervals along the island plateau, and left open overnight in 2010 and 2011’.

Line 159 - 160: Replace ‘the most common ones’ with ‘those’ and change ‘in GenBank’ to ‘on GenBank’. I appreciate that the authors have provided a table of sequences generated by DNA metabarcoding for pellets and Sanger sequencing for voucher specimens. However, I stand by my previous comment where I said that the authors should really have performed in silico analysis and reported the results.

Results should include a summary of which species present on the island were represented on GenBank/BOLD, which species were missing reference sequences, which species were barcoded in their study, and which species amplified with each primer set using a program that simulates PCR, e.g. ecoPCR. A species list for their study area, Selvagem Grande, could be constructed using public databases (e.g. GBIF, https://www.gbif.org/), based on previous studies or based on their own trapping efforts. In fact, the authors mention such a species list in Lines 224-225. With this species list, they can perform a GenBank/BOLD search to identify which species are represented and which species have no sequence records. They can highlight which gaps they filled through their own DNA reference database construction and which still remain for future studies to procure. They can download sequence data from GenBank/BOLD for species in their list and combine it with their own barcoding efforts to create a single fasta file to convert to an ecoPCR database. ecoPCR can then be performed for each primer set to identify which species should have amplified. Any additional species that PCR-amplified but did not amplify in silico will likely be species with reference sequences that do not include both the forward and reverse primer. Any species which amplified in silico but did not PCR-amplify would indicate amplification bias.

If the authors did not conduct an in silico analysis, they should provide justification for their primer choices. Why do they expect the selected primers to amplify taxa in their study area or more broadly, geographic region?

Lines 171-172: ‘following the criterion of the minimum number’ – I have not heard of this before. Could the authors add a sentence and briefly describe please?

Line 176-189: I suggest moving this section on sequencing of pellet samples below Lines 191-207 on reference database construction, or use subheadings to distinguish the two sections. Typically, you would construct your reference database before doing any DNA metabarcoding. Perhaps the authors did not do things in this order, but it is a more logical flow to the manuscript.

Lines 178-184: Change to ‘Three different DNA fragments were chosen to identify the distinct prey groups presumably preyed by the study species. For plants, the g/h primers that target the short P6-loop of the chloroplast trnL (UAA) intron were used (Taberlet et al., 2007). For invertebrates, modified IN16STK-1F/IN16STK-1R primers that target the mitochondrial 16S rRNA gene were used (Kartzinel & Pringle, 2015). For vertebrates, the 12sv5F and 12Ssv5R primers that target the V5-loop of the mitochondrial 12S rRNA gene (Riaz et al., 2011) were used (Tables S1, S2 and S3)’.

Line 193: Change ‘to allow the match with’ to ‘to match’ and ‘Standard’ to ‘The standard’.

Line 195: Change ‘confirming’ to ‘confirmation of’.

Line 210: Insert ‘the’ after ‘following’ and move ‘protocol’ after ‘Preparation’.

Line 213: Insert ‘,’ before ‘diluted’ and change ‘in’ to ‘on’.

Line 222: Insert ‘by’ before ‘comparing’.

Lines 225 - 227: Remove ‘of’ before ‘BLAST’. Reword as ‘Sequences with less than 90% BLAST identity were assigned to class level. Sequences with 90–95% BLAST identity were assigned to family level. Remaining sequences with 95% BLAST identity or higher were assigned to species or genus level’.

Line 227: ‘Prey items’ – I assumed the authors mean morphological prey items? Please clarify.

Line 233: Insert ‘assignments’ after ‘level’.

Lines 237-238: Reword as ‘This approach was not used for the 2017 samples as it is not possible to estimate the number of individuals belonging to each prey species using metabarcoding’.


Results

Lines 298-301: I actually interpret this as DNA metabarcoding is revealing hidden dietary diversity that the traditional sampling is not capturing, and that is why the accumulation curve does not reach a plateau. This is a main conclusion from the Discussion.


Discussion

Line 309: Insert ‘,’ after ‘beetles’ and after ‘biomass’.

Line 318: Insert ‘,’ after ‘May’.

Lines 322-325: I believe the authors are describing results from Hódar & Pleguezuelos (1999) here but this needs to be made clearer, e.g. ‘Hódar & Pleguezuelos (1999) showed that between April and July, …’.

Line 353: Change ‘notice’ to ‘note’ and ‘does not allow to obtain’ to ‘do not provide’.

Line 354: Change ‘impossible the detection of’ to ‘it impossible to detect’.

Line 356: Insert ‘always’ before ‘happen’.

Line 366: Remove ‘with’.

Line 367: Change ‘so in’ to ‘thus’.

Line 372: Change ‘fairly’ to ‘roughly’.

Line 374: Change ‘those’ to ‘these’.

Lines 374-377: Reword as ‘The small size of this gecko (average snout-vent length circa 6 cm) may prevent ingestion of large seeds or harder plant parts, which may explain the lack of plant material detected using morphological identification of Selvagens gecko pellets’.

Line 381: Change ‘ contains’ to ‘contents’.

Line 382-383: Change ‘Applying this method to’ to ‘Assessing stomach contents of’.

Line 386: Change ‘But’ to ‘However,’.

Line 387: Remove ‘could’.

Lines 393-394: Reword as ‘Due to the low number of terrestrial predators, it is common for insular reptile species to become diet generalists as they can reach high densities and face higher competition for food’.

Line 396: Change ‘of options, causing them to’ to ‘and even’.

Line 402: Change ‘of’ to ‘for’.

Line 407: Insert ‘,’ after ‘approach’.

Line 408: Remove ‘,’ after ‘OTUs’.

Line 409: Remove ‘case of’.

Line 412: Change ‘could show’ to ‘showed’.

Line 416: Change ‘could detect’ to ‘detected’.

Line 419: Change ‘evidences’ to ‘evidence,’.

Line 425: Insert ‘Selvagens gecko’ before ‘DNA’.

Line 428: Change ‘number’ to ‘range’.

Line 431: Move ‘only’ to come after ‘sampling’.

Line 433: Change ‘integrate information’ to ‘integrate the information they generate’. Insert ‘,’ after ‘generate’.

Line 465: Change ‘it mainly consists in’ to ‘diet mainly consisted of’.

Line 466: Remove ‘,’ after ‘feathers’.

Line 474: Insert ‘,’ after ‘limitations’.

Line 475: Remove ‘the’ before ‘successful’ and change ‘the range’ to ‘a range’.

Line 476-481: Reword as ‘DNA-based diet studies have been shown to describe the diet profile of a wide range of species with higher resolution and greater efficacy than classical methodologies, e.g. birds (Trevelline et al., 2016), bats (Hope et al., 2014) and sea lions (Hardy et al., 2017). Therefore, DNA metabarcoding is a valuable and revolutionary tool for conservation research and management (Allendorf & Luikart, 2009)’.

Line 484: Change ‘studies’ to ‘methods’.

Line 485: Change ‘species diet’ to ‘species’ diet’.

Lines 486-488: Reword as ‘Metabarcoding methods, even when deployed as extremely quick surveys, can deliver holistic results on diet composition, diversity, and ecological networks at relatively low cost (Lopes et al., 2019)’.

Line 489: Change ‘in’ to ‘to’ and ‘this insular species’ to ‘the insular Selvagens gecko’.

Lines 489-491: Remove line break to read as one paragraph and change ‘high’ to ‘large’.

Line 493: Remove ‘and’.

Line 494: Change ‘as it is the case of’ to ‘as is the case for’.

Line 499: Change ‘In this particular case,’ to ‘Based on our results,’.

Line 500: Insert ‘Selvagens’ before ‘gecko’.

Line 502-503: Change ‘the diet. Enlightening the referred matters’ to ‘its diet. Shedding light on these questions’ and ‘in’ to ‘for’.

Lines 504-505: Change ‘We hope our work has contributed to establishing guiding actions of future management planning’ to ‘Our work will contribute towards establishing guidelines for future management of the Selvagens gecko and its habitat’.

Table 1: Insert ‘,’ after ‘2017’ and ‘species’ before ‘sampled’.

Table 2: Change ‘Selvagem gecko’ to ‘the Selvagens gecko’.

Figure 2: Change ‘classical and metabarcoding methods’ to ‘morphological identification and DNA metabarcoding’.

Figure 3: Change ‘studies’ to ‘identification’.

---

## Round 0.3 · accepted · Accept

Congratulations again! Please work with our production team to get your paper published.